# An Opportunity to END TB: Using the Sustainable Development Goals for Action on Socio-Economic Determinants of TB in High Burden Countries in WHO South-East Asia and the Western Pacific Regions

**DOI:** 10.3390/tropicalmed5020101

**Published:** 2020-06-18

**Authors:** Srinath Satyanarayana, Pruthu Thekkur, Ajay M. V. Kumar, Yan Lin, Riitta A. Dlodlo, Mohammed Khogali, Rony Zachariah, Anthony David Harries

**Affiliations:** 1The Union South-East Asia (The USEA) Office, C-6 Qutub Institutional Area, New Delhi 110016, India; pruthu.tk@theunion.org (P.T.); akumar@theunion.org (A.M.V.K.); 2International Union against Tuberculosis and Lung Disease, 68 Boulevard Saint Michel, 75006 Paris, France; ylin@theunion.org (Y.L.); rdlodlo@theunion.org (R.A.D.); adharries@theunion.org (A.D.H.); 3Yenepoya Medical College, Yenepoya (Deemed to be University), University Road, Deralakatte, Mangalore 575018, India; 4International Union against Tuberculosis and Lung Disease, No.1 Xindong Road, Beijing 100600, China; 5Special Programme for Research and Training in Tropical Disease (TDR), World Health Organization, Avenue Appia 20, 1211 Geneva, Switzerland; khogalim@who.int (M.K.); zachariahr@who.int (R.Z.); 6London School of Hygiene and Tropical Medicine, Keppel Street, London WC1E 7HT, UK

**Keywords:** tuberculosis, End TB, sustainable development goals, South-East Asia, Western Pacific Region, national TB program, socio-economic determinants

## Abstract

The progress towards ending tuberculosis (TB) by 2035 is less than expected in 11 high TB burden countries in the World Health Organization South-East Asia and Western Pacific regions. Along with enhancing measures aimed at achieving universal access to quality-assured diagnosis, treatment and prevention services, massive efforts are needed to mitigate the prevalence of health-related risk factors, preferably through broader actions on the determinants of the “exposure-infection-disease-adverse outcome” spectrum. The aim of this manuscript is to describe the major socio-economic determinants of TB and to discuss how there are opportunities to address these determinants in an englobing manner under the United Nations Sustainable Development Goals (SDGs) framework. The national TB programs must identify stakeholders working on the other SDGs, develop mechanisms to collaborate with them and facilitate action on social-economic determinants in high TB burden geographical areas. Research (to determine the optimal mechanisms and impact of such collaborations) must be an integral part of this effort. We call upon stakeholders involved in achieving the SDGs and End TB targets to recognize that all goals are highly interlinked, and they need to combine and complement each other’s efforts to end TB and the determinants behind this disease.

## 1. Introduction

Tuberculosis (TB), caused by the bacteria *Mycobacterium tuberculosis* (Mtb), is one of the top 10 leading causes of death world-wide and the leading cause of death from a single infectious agent [1]. Mtb spreads from person to person through airborne droplet nuclei. When a person with active TB of the lungs or throat, coughs or sneezes, droplets containing Mtb are expelled into the air and inhalation of this contaminated air may cause TB infection [2]. Once infected, about 5–15% of the people develop active TB disease in their lifetime, with the risk of developing the disease being highest in the first two years of infection [3,4]. About 1.7 billion people or 23% of the world’s population are estimated to be infected with Mtb, of which 55.5 million (0.8% of the world’s population) are estimated to be recently infected and at high risk of progression to active TB [5]. The aim of this manuscript is to describe the major socio-economic determinants of TB and to discuss how there are opportunities to address these determinants in an englobing manner.

## 2. TB Burden: Globally and in the WHO South-East Asia and Western Pacific Regions

Globally, about 10 million people developed TB in 2018, with this number being relatively stable in recent years. The annual TB incidence rate ranged from less than 5 people per 100,000 population to more than 500 people per 100,000 population, with the global average of 130 people per 100,000 population. In 2018, an estimated 1.2 million HIV-negative people and an additional 251,000 HIV-positive people died from TB. Although these numbers are large, the number of deaths among HIV-negative people has reduced by 27% from 1.7 million in 2000 to 1.2 million in 2018. Similarly, the number of deaths among HIV-positive people has reduced by 60%, from 620,000 in 2000 to 251,000 in 2018.

Geographically, in 2018, the TB burden varied across the six regions of the World Health Organization (WHO): this comprised 44% in the South-East Asia region [6], 24% in the Africa region, 18% in the Western Pacific region, 8% in the Eastern Mediterranean region, and 3% each in the Americas and Europe. Globally, 30 high TB burden countries accounted for 87% of the world’s patients, with 11 of these being in the South-East Asia and Western Pacific regions (Table 1). Five countries in the South-East Asia and Western Pacific regions accounted for more than half of the global total: India—27%, China—9%, Indonesia—8%, the Philippines—6%, and Bangladesh—4%. Even within countries, the distribution of the TB burden is highly heterogeneous [7].

## 3. End TB Strategy

In 2014, to rid mankind of the enormous burden of TB, the 67th World Health Assembly adopted a resolution to make the world free of TB by the year 2035. WHO’s “End TB Strategy” provides a holistic overview of this resolution and has four principles and three pillars [8]. The three high-level target indicators of the End TB Strategy are reductions in TB deaths by 95%, reductions in the TB incidence rate by 90% and the percentage of TB patients and their households experiencing catastrophic costs being maintained at zero. These indicators and targets are relevant to all countries, with interim milestones to be achieved by 2020, 2025 and 2030.

### Progress towards the 2020 Milestones of the End TB Strategy

By the end of 2019, at the global level, most of the WHO regions and many high TB burden countries were not on track to reach the End TB Strategy’s 2020 milestones (20% reduction in TB incidence rate, 35% reduction in the number of TB deaths and reduction in the households experiencing catastrophic costs to 0%). The reduction in the cumulative global TB incidence rate between 2015 and 2018 was only 6.3%, and the reduction in the total number of TB deaths between 2015 and 2018 was 11% [6]. Table 2 shows the reduction in the TB incidence rate, TB mortality, and catastrophic costs in the 11 high TB burden countries in the Asia Pacific region (which comprises South-East Asia and the Western Pacific). However, with this rate of decline in incidence and mortality, and with the data on catastrophic costs unavailable, it is unlikely that any of the countries in the Asia Pacific region will be able to reach all the End TB Strategy’s 2020 milestones.

## 4. Factors Influencing the Risk of Exposure to Mycobacterium tuberculosis (Mtb), Infection, the Progression from Infection to Disease and Adverse Treatment Outcomes

There are two important aspects to understanding the TB epidemiology. First, Mtb infection in humans results in a spectrum of clinical presentations. As mentioned earlier, most infections are subclinical and asymptomatic, with Mtb replication contained by the host immunity—a condition called latent TB infection (LTBI)—and only a small subset of infected individuals presenting with symptomatic, active TB disease. Even within and between these two states, there is a wide ranging spectrum of Mtb bacterial load, immune responses, pathologies and clinical presentations [9]. Second, like all other infectious diseases, the risk of infection and disease is dependent on the characteristics and interaction of the bacteria, the human host and the environment [7]. A good understanding of these factors and their unique complex interactions—at both the population level and the individual level—is crucial for designing the intervention strategies to mitigate the TB burden.

The factors that influence the risk of exposure to Mtb, infection, the progression of infection to disease, and adverse treatment outcomes (such as death) are shown in Box 1 [10,11]. The factor that is essential for TB infection and disease is close contact with a person with a person with infectious TB disease; the greater the closeness, bacterial load and duration of contact, the higher the chances of infection. Other factors such as age, sex, tobacco use, alcohol use, malnutrition, human immunodeficiency virus (HIV) infection, diabetes mellitus and silicosis increase the risk of infection, the progression from infection to disease and adverse TB treatment outcomes, and are therefore called major health-related risk factors. Factors such as poverty, socio-economic and/or gender inequality, food and/or job insecurity and weak health systems affect not only all aspects of the “exposure-infection-disease-adverse outcome” spectrum, but also several aspects of the health of populations in general, and are therefore called the critical underlying ‘determinants’ or the ‘root causes’ of TB. While age and sex are not modifiable, all the other factors listed in Box 1 can be modified by human interventions. Table 3 shows the health-related risk factors and the corresponding lifetime increase in the risk of TB disease and the ‘population attributable fraction (PAF)’ of these factors [12,13,14,15,16,17,18].

Box 1Factors influencing the “exposure-infection-disease-adverse outcome” spectrum of TB.
**The essential factor for TB infection and TB disease**
Close contact with a person with infectious TB disease
**Major health-related risk factors (factors that increase the chances of infection, disease, adverse TB treatment outcomes)**
Age, sex, tobacco use, alcohol abuse, malnutrition, HIV infection, diabetes mellitus, exposure to indoor air pollution, silicosis, intake of immunosuppressive drugs/medications (e.g., tumor necrosis factor-alpha (TNF) antagonists, corticosteroids)
**Major underlying determinants**
Poverty, socio-economic and gender inequality, overcrowding, food and job insecurity, weak health systems

### 4.1. Relationship between Various Health-Related Risk factors and Major Underlying Socio-Economic Determinants of TB

In the past, the trends in TB incidence rates in high-income countries clearly show that efforts towards improving the socio-economic status, living conditions and nutritional status (as was seen before and soon after the world wars) resulted in the rapid decline in the TB burden, and, deterioration in these conditions (during times of war) increased the TB incidence rates. Both in high and low-income countries, TB predominantly affects people of lower socio-economic status [19,20].

Most of the risk factors for TB are associated with poverty, socio-economic and gender inequalities, and living conditions [21]. Malnutrition, poor housing/living conditions, and overcrowding are direct markers of poverty [22]. People from lower socio-economic groups are more likely to live and/or work in overcrowded settings, experience greater food insecurity, have lower levels of awareness about healthy behavior, and are less likely to have access to quality health care services [23]. They are also more likely to come into contact with people with active TB disease. The prevalence of tobacco use, alcohol use, HIV, and diabetes mellitus is relatively higher in people of low socio-economic status groups in various settings [24,25,26,27,28].

The Multisectoral Accountability Framework to accelerate progress to end TB (MAF-TB) by 2035 developed by the WHO in 2019 [29] urges governments to address a wide range of socio-economic determinants through collaborations and partnerships. Although the primary responsibility to pursue public health in all policies rests with different ministries within governments, the national TB programs, as champions and implementers of the TB care and prevention services in the countries, should take the lead in developing partnerships and support the implementation of the multisectoral accountability framework, both through advocacy and by helping to address the social conditions of patients and their families.

### 4.2. Need for Action on Risk Factors and Major Determinants of TB

TB (as described earlier) is a multifactorial disease, and the achievement of the End TB Strategy milestones requires universal access to quality-assured diagnosis, treatment, and prevention services promptly. For this, it is necessary to strengthen the national TB programs, by ensuring adequate resources for deploying latest WHO-endorsed rapid TB diagnostics and drug susceptibility testing (DST) facilities, the provision of appropriate treatment services for drug-susceptible and drug-resistant TB, preventive treatment for high risk individuals (people living with HIV and household and other close contacts of TB patients), and the implementation of infection control measures in all health facilities. While these are necessary, it is being increasingly recognized that End TB targets are ambitious and unlikely to be achieved by these measures alone [10,11,30]. Massive efforts are needed to mitigate the prevalence of health-related risk factors, preferably through broader actions on the determinants of the “exposure-infection-disease-adverse outcome” spectrum, such as health system strengthening, poverty alleviation, addressing socio-economic and gender inequality, limiting job loss and food insecurity, improving housing quality and reducing overcrowding. An increase in the health-related risk factors of TB or the worsening of the determinants of TB (as is currently happening due to the socio-economic consequences of the SARS-CoV-2 pandemic [31]) can harm the global progress made towards ending TB.

## 5. Framework for Action on Social Determinants of Tuberculosis: Sustainable Development Goals

The sustainable development goals (SDGs) [32] are a list of 17 global goals, which outline the vision, principles, and commitments of all United Nations member countries to a fairer and more sustainable world, now and in the future. The SDGs, launched in 2015 by the United Nations General Assembly, and intended to be achieved by the year 2030, are part of UN Resolution 70/1, the 2030 Agenda. The 17 SDGs are mentioned in Box 2.

Box 2The Sustainable Development Goals (2015 to 2030) *.
**No Poverty**

**Zero Hunger**

**Good Health and Well-being**

**Quality Education**

**Gender Equality**

**Clean Water and Sanitation**

**Affordable and Clean Energy**

**Decent Work and Economic Growth**
Industry, Innovation, and Infrastructure 
**Reducing Inequality**

**Sustainable Cities and Communities**
Responsible Consumption and Production
**Climate Action**
Life Below WaterLife on LandPeace, Justice, and Strong InstitutionsPartnerships for the Goals

The seventeen SDGs each have a list of targets that are measured with indicators [33]. These goals necessitate collaboration and the alignment of all actions to secure a fair, healthy, and prosperous future for everyone on this planet—Earth. These SDGs provide a framework for action on the determinants of TB disease. Action on the determinants of TB through the SDGs framework will also mean endorsing the socio-ecological model of health which outlines that disease prevention and its mitigation may require action at five levels: individual, interpersonal, organisational, community and public policy [34] and addressing every TB determinant may require actions at these five levels.

### 5.1. Sustainable Development Goals that Are Likely to Have a Significant Impact on the Burden of TB

#### 5.1.1. SDG Goal 1: No Poverty

As discussed above, poverty is a significant determinant of several aspects of health including TB [22,35]. Globally, more than 90% of TB cases and deaths occur in low and middle income countries [36]. There is an inverse correlation between a country’s gross domestic product (GDP) per capita and TB incidence rates [21]. Therefore, efforts to reduce poverty will have a substantial impact on the TB burden. Despite great progress made globally in reducing poverty levels, some countries in the WHO South-East Asia and Western Pacific Regions, such as India (~21%) and Papua New Guinea (38%), have high reported levels of people living below the international poverty line (SDG Indicator 1.1.1 in Table (SDG Indicator 2.1.1 in Table 4).

#### 5.1.2. SDG Goal 2: Zero Hunger

Hunger leads to undernutrition, which is one of the significant determinants of TB [37]. It is estimated that in India (the highest TB burden country in the world), where the prevalence of undernutrition is high, nearly 50% of TB cases are attributable to undernutrition [38,39]. Undernutrition is also ubiquitous in all high TB burden countries in Asia and the Pacific region (SDG Indicator 2.1.1 in Table 4). Therefore, ending hunger and improving the nutritional status of populations can dramatically reduce the burden of TB.

#### 5.1.3. SDG Goal 3: Good Health and Well Being

One of the key targets of this goal (Target 8) is to achieve universal health coverage (UHC) by 2030 [40]. UHC means that everyone can access and receive sufficient and quality health services that they need without suffering financial hardship. There are two key indicators to monitor progress. They are: a) UHC service coverage index (SCI)—SDG Indicator 3.8.1 (Table 5), and b) the percentage of the population experiencing expenditures on health care that are relatively large in relation to the household expenditures or income—SDG Indicator 3.8.2 (Table 5). The achievement of UHC and improved patient-centered TB care will have a direct effect on the reach and delivery of quality TB services and on catastrophic costs incurred by TB patients and their families [40]. Apart from this, the SDG goal also has indicators for dramatically reducing the prevalence of HIV, tobacco and alcohol use, and diabetes mellitus, all of which will have a considerable impact on reducing the TB burden.

#### 5.1.4. SDG Goal 4: Quality Education

Quality education typically leads to better and secure jobs, more money and higher purchasing power, resulting in better access to quality health care (including for TB) [41]. Higher earnings also allow people to afford homes in safer neighbor hoods, as well as consume healthier diets. Incorporating health within the ambit of ‘quality education’ builds knowledge, skills, and positive attitudes about health and all other determinants of health, which can directly or indirectly have a considerable impact on efforts to End TB [42]. Education also improves the ability to identify the symptoms suggestive of TB and seek timely care for diagnosis and treatment of TB [43], thus, limiting the delays in the diagnosis of TB and the community spread of the disease.

#### 5.1.5. SDG Goal 5: Gender Equality

TB can affect either gender. In almost all countries, the notification rates of TB are higher in males than in females. Although more men than women develop TB disease and die from it, TB is nevertheless a leading infectious cause of death among women. Higher tuberculosis notification rates in men partly reflect epidemiological and biological differences in exposure, risk of infection, and progression from infection to disease [44]. Despite this, in several countries, gender inequality, socio-economic, and cultural factors act as barriers to accessing health care among women. These may lead to the under-detection and under-notification of TB in women. The stigma and discrimination associated with TB and certain co-morbidities, such as HIV infection, adversely affect women more than men, often leaving them in a more vulnerable position [45,46]. It is also widely believed that the medical care-seeking behavior of men and women with TB is mostly determined by how they and those around them perceive the symptoms, regard the diagnosis, accept the treatment, and complete it. Gender may influence each of these components and affect the early detection of the disease and its outcome [47]. Studies that have assessed gender differences have shown that, on average, women are either undiagnosed, or diagnosed late in the course of TB disease when compared to men [48,49]. Promoting gender equality in all spheres of life helps to mitigate some of these issues and contribute towards ending TB.

#### 5.1.6. SDG Goal 6: Clean Water and Sanitation

Access to clean water and sanitation is essential to reduce illness, malnutrition, poor physical and cognitive development, and death due to water-borne diseases [50]. In countries of the Asia-Pacific region, the proportion with access to clean water ranges from 16.7% in rural Cambodia to 92.3% in urban China. The proportion with access to safe sanitation ranges from 5.1% in rural Democratic People’s Republic of Korea to 83.7% in urban China. The provision of clean water and sanitation can affect the TB burden by reducing infections and improving nutritional status.

#### 5.1.7. SDG Goal 7: Affordable and Clean Energy

There is evidence of an association between indoor air pollution (such as that caused by the burning of solid fuels for cooking at homes), outdoor ambient air pollution and TB infection and TB disease [51,52,53,54]. Depending on the prevalence of indoor air pollution, the fraction of TB cases attributable to indoor air pollution varies across countries. The proportion of the population using clean fuels in the South-East Asia and Western Pacific regions ranges from 11% in DPR Korea to 74% in Thailand (Table 4; indicator 7.1.2). Interventions such as clean cook stoves to reduce the adverse effects of indoor air pollution merit rigorous evaluation [50], particularly in high TB burden countries in Asia and the Pacific, where the prevalence of both indoor air pollution and TB is high. Clean energy is also expected to reduce outdoor air pollution and improve ambient air quality, which can reduce the risk of TB [55].

#### 5.1.8. SDG Goal 8: Decent Work and Economic Growth

TB predominantly affects people in the economically productive age-groups [56]. Apart from providing stable and regular job opportunities for the economically productive age groups, provision for early diagnosis and treatment of TB at workplaces, making workplaces safe by reducing the chances of airborne transmission of infections, adopting favorable policies towards social/job security in case of diseases and reducing occupational diseases like silicosis, will help in reducing the TB burden and improving the economic productivity of the workforce [57]. Stable/formal employment also increases access to employer-sponsored social health insurance programs and paid sick leaves, both of which are known to be associated with the reduced risk of occupational diseases and all-cause mortality [58,59,60].

#### 5.1.9. SDG Goal 10: Reduced Inequalities

Empowering and promoting social, economic and political inclusion of all, irrespective of age, sex, sexual orientation, disability, race, ethnicity, origin, religion or economic or another status will help in mitigating the effects of socio-economic disparities—disparities that are key drivers of all the risk factors of TB [61].

#### 5.1.10. SDG Goal 11: Sustainable Cities and Communities

Rapid industrialization, urbanization, and migration—dominant occurrences in most developing countries in the Asia Pacific region—can create ideal conditions for infectious diseases (including TB) to flourish [62,63], unless accompanied by proper urban planning, social reforms, environmental protection, adequate housing, transportation, and a well-coordinated and robust health system.

#### 5.1.11. SDG Goal 13: Climate Action

Climate change that manifests itself in the form of higher variations in temperatures and rainfall is known to have a substantial effect on several aspects of human health and behaviour (such as crowding, migration, changes in food habits), either directly or through several intermediaries, resulting in an increase in the burden of infectious diseases including TB [64,65].

The linkages between the various SDGs and TB are pictorially depicted in Figure 1.

## 6. Role of National TB Programs in Accelerating the Progress towards Achieving SDGs

Together, TB and poverty form a vicious cycle: TB decreases people’s capacity to work and adds to treatment expenses. This, in turn, exacerbates their poverty. Poor people also go hungry and live in close, unhygienic quarters, where TB and its risk factors flourish. Progress in ending TB will accelerate the progress on SDG goal 1, and through it, progress on other related SDGs [36].

According to the WHO’s “A Multisectoral Accountability Framework to accelerate progress to end Tuberculosis by 2030 (MAF-TB)” [29], the following are recommended actions for national TB programs:Development of national (and local) strategic and operational plans to end (or eliminate) TB, with a multisectoral perspective involving government and partners, consistent with the End TB Strategy.Development and use of a national MAF-TB.Establishment, strengthening or maintenance of a national multisectoral mechanism (e.g., inter-ministerial commission), tasked with providing oversight, coordination and a periodic review of the national tuberculosis response.Implementation of multisectoral actions on the social determinants of tuberculosis.Revisions to plans and policies, and associated activities, based on monitoring, reporting, and recommendations from reviews.

WHO has identified the following fourteen SDG indicators as relevant to TB. These include

Seven indicators relevant to TB under SDG 1, 2, 7, 8, 10 and 11 and seven indicators within SDG-3.
(1)SDG 1 (No poverty)—Indicator 1.1.1: Proportion of the population living below the international poverty line(2)SDG 1 (No poverty)—Indicator 1.3.1: Proportion of the population covered by social protection floors or systems(3)SDG 2 (Zero hunger)—Indicator 2.1.1: Prevalence of undernourishment(4)SDG 7 (Affordable and clean energy)—Indicator 7.1.2: Proportion of the population with primary reliance on clean fuels and technology(5)SDG 8 (Decent work and economic growth)—Indicator: 8.1.1: Gross domestic product (GDP) per capita(6)SDG 10 (Reduced inequalities)—Indicator: 10.1.1: Gini index for income inequality(7)SDG 11 (Sustainable Cities and Communities)—Indicator: 11.1.1: Proportion of the urban population living in slums

The levels of these seven indicators under SDG 1, 2, 7, 8, 10 and 11 at the end of 2018 in the 11 high TB burden countries in WHO South-East Asia and Western Pacific Regions are given in Table 4.

B.Seven indicators relevant to TB within SDG 3 are as follows:(1)SDG Indicator 3.8.1: Coverage of essential health services (UHC) measure with an UHC index based essential health services and ranges between 0 and 100(2)SDG Indicator 3.8.2: Proportion of the population with large household expenditures on health, as a share of total household expenditure or income(3)SDG Indicator 3.c.1: Current health expenditure per capita in current international dollars(4)SDG Indicator 3.3.1: Prevalence of HIV(5)SDG Indicator 3.a.1: Prevalence of smoking(6)SDG Indicator 3.4.1: Prevalence of diabetes(7)SDG Indicator 3.5.2: Prevalence of alcohol use disorder

The levels of these seven within SDG-3 indicators at the end of 2018 is in the 11 high TB burden countries in WHO South-East Asia, and Western Pacific Regions is given in Table 5.

National TB programs, Ministries of Health and Governments in the 11 high TB burden countries must realize that interventions beyond diagnosis and treatment of TB and LTBI are needed to reduce the risk factors and determinants of TB. TB programs must take a lead and list the stakeholders in their countries who are working on the other SDGs and equip themselves with resources and the necessary skillsets to engage with them. Since TB is not homogenously distributed within the country, there will be geographical areas/communities with high TB burdens. As a first step, TB programs should share details/information of high TB burden geographical areas, assess the socio-economic determinants that are locally relevant/prevalent, and facilitate concentrated action on those socio-economic determinants as a priority in these areas wherever possible. The list of sample interventions that can be undertaken to reduce the prevalence of the socio-economic determinants of TB (based on needs assessment) are given in Table 6. (In this table, using India as an example, we have highlighted public programs [69,70,71,72,73,74,75,76,77,78,79,80,81,82] that can be galvanized to address the determinants at the local level.)

The stakeholders/partners who can be involved in carrying out the interventions could be respective government departments (who have the mandate and jurisdiction to perform the intervention), non-governmental/community based organizations, private sector, developmental partners etc., who may share the common vision of improving the lives of the people. Identifying suitable partners will help the national TB programs, in facilitating their interventions in such geographic high TB burden areas. This holistic approach may contribute towards ending TB in such communities or geographic areas in a sustainable manner. Demonstration projects on how to operationalize intersectoral coordination and build partnerships (at the local level) are urgently needed to generate evidence and to show that the impact of such initiatives goes beyond the simple sum of its immediate returns. Research—to find out the optimal mechanisms for collaboration and to measure the impact of such collaborations—must also be undertaken, so that the lessons learnt are disseminated widely. Apart from this, national TB programs must also find themselves represented in all in-country developmental committees and agendas of the government, so that ending TB is seen, not just as a responsibility of the health sector, but seen as a necessity for human development in all spheres of life. International technical and funding agencies/organizations, like WHO, the Global Fund, etc., should advocate for progress on broader SDGs and provide technical and financial assistance on this aspect to TB programs.

## 7. Conclusions

Achieving End TB targets with the current pace of progress is highly challenging and un-realistic if the thrust is mainly medical and focusing only on ‘diagnosis and treatment’ of TB and LTBI, without addressing the underlying determinants of TB. While the strengthening of national TB programs under the framework of universal health coverage is quintessential for accelerating the progress towards End TB targets, the SDG framework provides an excellent opportunity for acting on several determinants and risk factors of TB. All stakeholders, be it government ministries, non-governmental organizations or private sectors involved in achieving SDGs and the End TB targets must recognize that most of their goals are strongly interlinked. Failure to acknowledge this fact may result in ineffective and inappropriate actions and a delay in the achievement of both SDG goals and End TB targets.

## Figures and Tables

**Figure 1 tropicalmed-05-00101-f001:**
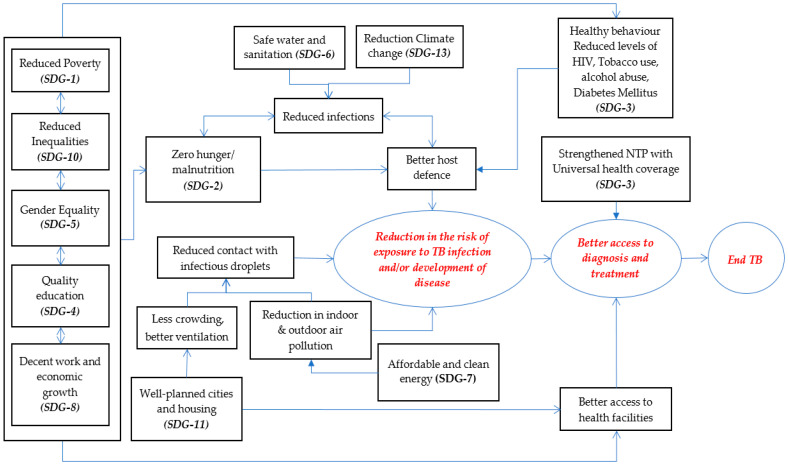
The possible direct and indirect linkages between SGDs 1, 2, 3, 4, 5, 6, 7, 8, 10, 11, 13 and reduction in tuberculosis burden. The arrows indicate the probable direction of action, with the bi-directional arrow indicating that effects in both the directions are possible. The relationships/pathways shown in this figure are ‘indicative’ only and not ‘definitive’. The size of the square boxes in the figure have been arrived at using the best fit function, and therefore the varying sizes or shapes of the text boxes/circles in the figure do not carry any special significance. SDGs 1, 4, 5, 8, 10 are grouped together, to indicate the author’s view that they are interdependent and can act alone or in a combined manner to influence the other aspects in the pathway.

**Table 1 tropicalmed-05-00101-t001:** List of high TB burden countries in the WHO South-East Asia and Western Pacific Regions.

Country	Annual TB Incidence in Thousands (Uncertainty Intervals)	The Approximate Annual Number of TB Deaths in Thousands (Best Estimates)
Bangladesh	357 (260–469)	47.2
Cambodia	49 (27–77)	6.7
China	866 (740–1000)	39.4
DPR Korea	131 (114–149)	20.0
India	2690 (1840–3700)	449.7
Indonesia	845 (770–923)	98.3
Myanmar	181 (119–256)	24.7
Papua New Guinea	37 (30–45)	4.7
Philippines	591 (332–924)	26.6
Thailand	106 (81–136)	11.5
Vietnam	174 (111–251)	13.2

Source: WHO Global TB Report 2019 [6].

**Table 2 tropicalmed-05-00101-t002:** High TB burden countries in the WHO South-East Asia and Western Pacific Regions and the relative change in crucial indicators between 2015 and 2018.

Country	TB Incidence Rate (Per 100,000 Population)			Number of TB Deaths (in Thousands)			Proportion of TB Patients Experiencing Catastrophic Costs (in 2018)
	2015	2018	Reduction *	2015	2018	Reduction *	
Bangladesh	221	221	0%	66.0	47.0	29%	NA
Cambodia	367	302	18%	3.8	3.4	11%	NA
China	65	61	6%	43.0	40.0	7%	NA
DPR Korea	513	513	0%	10.0	20.0	−100%	NA
India	217	199	8%	470.0	449.0	4%	NA
Indonesia	325	316	3%	102.0	98.0	4%	NA
Myanmar	391	338	14%	36.0	25.0	31%	NA
Papua New Guinea	432	432	0%	4.3	4.7	−9%	NA
Philippines	550	554	−1%	28.0	26.0	7%	NA
Thailand	163	153	6%	15.0	11.0	27%	NA
Vietnam	199	182	9%	17.0	13.0	24%	NA

Source: Point estimates from WHO Global TB Report 2019 [6]; NA = Data Not available; * negative sign indicates an increase in the TB incidence rate or number of TB deaths.

**Table 3 tropicalmed-05-00101-t003:** Health-related risk factors and the corresponding lifetime increase in the risk of TB disease and their corresponding population attributable fraction.

Risk Factor	Relative Risk of TB (95% Confidence Intervals) *	Estimated Prevalence of Risk Factor in the South-East Asia and Pacific Region **	Estimated Population Attributable Fraction (PAF) ***
HIV	19 (16–22)	0.2% [12]	3.5% (2.9–4.0%)
Alcohol abuse	3.3 (2.1–5.2)	13.5% [13]	23.7% (12.9–36.2%)
Undernourishment	3.2 (3.1–3.3)	11.5% [14]	20.2% (19.5–20.9%)
Smoking	1.6 (1.2–2.1)	24.8% [15]	13.0% (4.7–21.4%)
Diabetes Mellitus	1.5 (1.3–1.8)	8.6% [16]	4.1% (2.5–6.4%)
Indoor Air Pollution	2.0 (1.2–3.2)	38.6% [17]	27.8% (7.2–45.9%)

* Source: Global TB report 2019 [6]); ** these are approximates and vary widely across countries; *** PAF indicates the proportion of all cases of a particular disease in a population that is attributable to a specific exposure and is estimated based on the relative risk and the prevalence of the risk factor in the population [18].

**Table 4 tropicalmed-05-00101-t004:** Current levels of TB relevant SDG 1, 2, 7, 8, 10, 11 indicators in high TB burden countries in the WHO South-East Asia and Western Pacific Regions (in 2018).

Country	SDG Indicators						
	1.1.1 Proportion of the Population Living below the International Poverty Line	1.3.1 Proportion of the Population Covered by Social Protection Floors or Systems	2.1.1 Prevalence of Undernourishment	7.1.2 Proportion of the Population with Primary Reliance on Clean Fuels and Technology	8.1.1 Gross Domestic Product (GDP) Per Capita, PPP (Constant 2011 International Dollars)	10.1.1 Gini Index for Income Inequality *	11.1.1 Proportion of the Urban Population Living in Slums
Bangladesh	15%	18%	15%	18%	3500	32	55%
Cambodia	NA	3.1%	18%	18%	3700	NA	55%
China	0.7%	63%	8.7%	59%	15,300	39	25%
DPR Korea	NA	NA	43%	11%	NA	NA	NA
India	21%	30%	15%	41%	6500	36	24%
Indonesia	5.7%	57%	7.7%	58%	11,200	38	22%
Myanmar	6.4%	2.3%	10%	18%	5600	38	41%
Papua New Guinea	38%	4.2%	NA	13%	3800	48	NA
Philippines	NA	41%	14%	43%	7600	40	38%
Thailand	0%	79%	9%	74%	16,300	36	25%
Vietnam	2%	35%	11%	67%	6200	35	27%

PPP = Purchasing Power Parity; NA = Not available; Source of data: [66,67,68]; * Gini index = measure of income distribution across income percentiles in a population. It ranges from 0% to 100%, with 0% representing perfect equality and 100% representing perfect inequality.

**Table 5 tropicalmed-05-00101-t005:** Estimated prevalence of TB relevant SDG-3 indicators in high TB burden countries in the WHO South-East Asia and Western Pacific Regions (in 2018).

Country	SDG-3 Indicators									
	3.8.1 Coverage of Essential Health Services (UHC) Measure with an UHC Index Based Essential Health Services and Ranges between 0 and 100	3.8.2 Proportion of the Population with Large Household Expenditures on Health as a Share of Total Household Expenditure or Income	3.c.1 Current Health Expenditure Per Capita in Current International Dollars	3.3.1 Prevalence of HIV	3.a.1 Prevalence of Smoking		3.4.1 Prevalence of Diabetes		3.5.2 Prevalence of Alcohol Use Disorder	
					Male	Female	Male	Female	Male	Female
Bangladesh	48	25%	91	0.1%	45%	1%	10%	9.3%	1.4%	0.3%
Cambodia	60	15%	229	0.5%	34%	2%	7.4%	6.9%	8.7%	1.8%
China	79	20%	761	NA	48%	1.9%	9.9%	7.6%	8.4%	0.2%
DPR Korea	71	NA	NA	NA	NA	NA	5.8%	5.9%	6.2%	1.0%
India	55	17%	241	0.2%	21%	1.9%	9.1%	8.3%	9.1%	0.5%
Indonesia	57	2.7%	363	0.4%	76%	2.8%	7.6%	8.0%	1.4%	0.3%
Myanmar	61	14%	291	0.7%	35%	6.3%	6.9%	7.9%	3.2%	0.6%
Papua New Guinea	40	NA	92	0.9%	49%	24%	15%	14%	8.8%	1.8%
Philippines	61	6.3%	342	0.1%	41%	7.8%	7.1%	7.3%	8.8%	1.8%
Thailand	80	2.2%	635	1.1%	39%	1.9%	8.3%	8.8%	10%	0.9%
Vietnam	75	9.4%	356	0.3%	46%	1.0%	5.5%	5.1%	9.8%	1.2%

NA = Not available; Source of data: [66,67,68].

**Table 6 tropicalmed-05-00101-t006:** Sample policies/interventions that can be implemented at the community level in high TB burden geographical areas within a country by the National TB program and its partners to address determinants of tuberculosis.

Determinant of TB	Sample Community Level Policies/Interventions to Address the Determinant	Example: Public Programs in India that Can Be Galvanized to Address the Determinant at the Local Level
To reduce poverty (SDG-1)	Skills building programs Providing access to credit/microfinance and markets Generating employment opportunities	Pradhan Mantri Kaushal Vikas Yojana (PMKVY)—A skill development scheme [69] Mahatma Gandhi National Rural Employment Guarantee Act (MGNREGA) 2005 (India)—Aims to enhance livelihood security in rural areas [70]
To reduce hunger (SDG-2)	Nutrition education Food subsidy interventions/ Cash transfers Targeted Supplemental nutrition assistance programs Finding ways to increase agricultural production and cost of food crops	National Food for Work Program (India) School mid-day meals program [75]
Improve the quality of education (SDG-4)	Improving physical infrastructure of schools (including sanitation facilities), providing teaching and learning materials, and training and hiring extra teachers. Enhancing community ownership and engagement in education Adult literacy, numeracy and education programs and special education programs to children and youth with disabilities and learning difficulties Compulsory and free education programs for children and youth	National Education Mission (Samagra Shiksha Abhiyan) [76]
Gender equality (SDG-5)	Awareness-raising and sensitization programs to reduce gender inequality, raise awareness about different gender identities and reduce gender-based violence Engagement with political leaders, religious leaders, community leaders to address practices that impair the rights of women and girls Empowering young girls and women by creating employment and education opportunities for their holistic growth and development Provide legal and psychosocial support services to address gender discrimination and human rights violations	Beti Bachao Beti Padhao Scheme (To provide education to girls’ and their welfare. To prevent the violation in the interest of girls. To celebrate the birth of a girl child.) [77] Support to Training and Employment Program for Women (STEP) [78] Mahila E-Haat (To help women to make financial and economic choices which will enable them to be a part of ‘Make in India’ and ‘Stand Up India’ initiatives) [79]
Clean water and sanitation (SDG-6)	Construction of toilets and latrines in all households and public spaces that flush into a sewer or safe enclosure. Information, Education and Communication to promote good hygiene habits (e.g., hand washing). Rainwater harvesting to collect and store rainwater for drinking or recharging underground aquifers. Provide low cost home water-treatment capability through the use of filters, chlorine tablets, plastic bottles for solar disinfection, or flocculants, to make drinking water safe. Invest in public piped water supply systems that provide good quality and potable water particularly for the poor.	Swachh Bharat Abhiyan-gramin (to eliminate open defecation and improve solid waste management in rural areas) [80] National Water Supply and Sanitation Program in India [81] National Rural Drinking Water Program (to provide safe and adequate water for drinking, cooking and other domestic needs to every rural person on a sustainable basis) [82]
Affordable and clean energy (SDG-7)	Provision of improved cookstoves, cleaner and drier fuels which aim to burn fuel more efficiently and therefore produce fewer harmful combustion products; Improving natural and artificial ventilation, to avoid air pollution inside the household; Changing cooking behavior and patterns, to reduce the amount of time an individual spends in proximity to a fire or stove; Altering regulatory or financial policies, with intent to improve access to advanced cookstoves or fuels and provide incentives for changes within communities or towards community development. Strict industrial regulations to control environmental air pollution	Pradhan Mantri Ujjwala Yojana (to distribute LPG gas connections to women of below poverty line families) [71]
Decent work and economic growth (SDG-8)	Encouragement for locally available micro, small, and medium sized enterprises through access to necessary financial services Generate employment opportunities Trainings/mentorship support for entrepreneurship Low interest loan schemes	Entrepreneurship Development Program (EDP) [72] Mahatma Gandhi National Rural Employment Guarantee Act (MGNREGA) 2005 (India) [70]
Reduced inequalities (SDG-10)	Local programs aimed at financial inclusion and social security or linkages to existing social security schemes	Jan Dhan-Aadhaar-Mobile programs (For Financial Inclusion to ensure access to financial services, namely banking savings and deposit accounts, remittance, credit, insurance and pension in an affordable manner and to prevent leakage of government subsidies) [73]
Sustainable cities and communities (SDG-11)	Local implementation of slum development schemes/ slums upgrading programs which includes improvements in housing conditions, water supply and sanitation, roads, ground stabilization, storm water drainage etc.,	Jawaharlal Nehru National Urban Renewal Mission (a city-modernization scheme) [74]
Climate action (SDG-13)	Efforts to reduce carbon emissions through increased generation of power using renewable sources of energy Increase additional forest and tree cover	National Solar mission [83] National Afforestation Program [84]

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
