# Peer review of "An Opportunity to END TB: Using the Sustainable Development Goals for Action on Socio-Economic Determinants of TB in High Burden Countries in WHO South-East Asia and the Western Pacific Regions"

_tropicalmed, 2020, doi:10.3390/tropicalmed5020101_

Round 1

Reviewer 1 Report

This is an ambitious, comprehensive, and impressive effort that brings important issues to the forefront. This manuscript identifies major socioeconomic determinants of TB and how these can be addressed under the United Nations Sustainable Development Goals (SDGs) framework. While this reviewer acknowledges the importance of these broad global goals towards achieving a fairer and more sustainable world, much of this work towards these targets may be challenging for many of these high burden TB countries to address simultaneously. It seems like this article could almost be written with any health outcome as its focus given that the SDGs are so cross-cutting and universally needed.

The authors put the responsibility on the National TB Programs to work with stakeholders on the SDGs and to develop mechanisms to engage with them to facilitate action on 28 social-economic determinants in high TB burden geographical areas. This seems like it should be the responsibility of larger government entities or ministries of health. It seems like a tall and unrealistic order for many National TB Programs (NTPs) that may be under resourced to begin with and should focus first and foremost on achieving excellent TB disease diagnostic, treatment, and prevention services. Although the authors give some acknowledgment to responsibility of governments and ministries of health (lines 130-133), it is unclear that they limit the role of NTPs to advocacy only. This should come sooner in the paper. There should also be a specific call out to the responsibilities of the governments and ministries of health, as well and larger organizations like WHO, Global Fund, etc. to advocate for progress on the broader SDGs.

In addition to acknowledging the high-level indicators of the End TB Strategy (e.g., reductions in TB deaths by 95%, reductions in the TB incidence rate by 90% and the percentage of TB patients and their households experiencing catastrophic costs being maintained at zero) there needs to be some attention given to the essential and known programs and policies necessary to achieve these indicators. There should an acknowledgement of essential core TB services and strategies that should be the foundation for any successful NTP. Perhaps the authors should acknowledge and mention what essential services TB programs should include (https://www.who.int/tb/publications/2015/end_tb_essential.pdf?ua=1). A few key principles of successful NTPs , including, but not limited to the following examples, should be mentioned:

  • Early diagnosis and prompt treatment of all persons of all ages with any form of drug susceptible or drug-resistant TB is fundamental. WHO-endorsed rapid TB diagnostics and drug susceptibility testing (DST) should be available to all who need it and prioritized for persons at risk of multidrug-resistant TB (MDR-TB) and HIV-associated TB.
  • Appropriate treatment of drug-susceptible and drug-resistant TB should be available and accessible to all who need it. Proper drug safety monitoring and management should be pursued. All relevant care providers should be engaged in the delivery of TB care.
  • Joint TB and HIV programming should be pursued for integrated and decentralized delivery of services for TB and HIV; Close contacts of people with TB, people living with HIV (PLHIV) and workers exposed to silica dust should be systematically screened for TB and considered for the treatment of latent TB infection (preventive therapy).
  • TB infection control measures should be applied in all settings.

The authors should acknowledge that it is essential to have a strong NTP and TB strategy in a country as a foundation to see progress. It would be helpful to highlight one or two countries that are exemplary. What are some of the reasons that some countries are on the right path? For example, the authors acknowledge that Cambodia and Myanmar may achieve the End TB Strategy for 2020 Milestons. Why? Also, in addition to acknowledging country specific successes, it would be important to acknowledge larger TB prevention and control strategies such as expanded TB preventive Treatment (TPT) among HIV-infected individuals.

The authors may also want to acknowledge the Social-Ecological Model of health which encourages us to move beyond a focus on individual behavior and toward an understanding of the wide range of factors that influence health outcomes. It has an emphasis on multiple levels of influence (such as individual, interpersonal, organizational, community and public policy) and the idea that behaviors both shape and are shaped by the social environment. Many of the SDGs the authors are advocating to influence are in the organizational, community, and public policy spheres. They should acknowledge this literature and disease-area examples using this model to explain and influence health outcomes.

In Box 1, authors may want to acknowledge medical treatments that suppress the immune system (such as tumor necrosis factor-alpha (TNF)antagonists, corticosteroids, or drug therapy following organ transplants).

Below, please find feedback on the specific SDGs as presented in the manuscript:

  • It would seem that the authors would want to give additional attention to the SDG Goal 3: Good Health and well being and the discussion of universal health care coverage, given it is likely the most significant SDG relative to TB prevention and control and other diseases. There must be additional literature that could be incorporated into the discussion of this goal. Is there any modeling literature, specific to TB that can be mentioned?
  • For SDG Goal 7: Sustainable and clean energy the authors focus almost primarily on indoor air pollution. There should be some additional acknowledgment and discussion of outdoor air pollution and its impact on diseases, in particular respiratory disease.
  • For SDG Goal 8: Decent work and economic growth the authors may also want to acknowledge and discuss that gainful employment may provide access to health insurance or workplace health programs in some settings/countries.
  • For SDG Goal 10: Reduced inequalities consider adding sexual orientation in the list of factors listed.
  • For SDG Goal 11: Sustainable Cities and Communities consider adding adequate housing and transportation
  • SDG Goal 13: Climate Action discussion is too general and not specific to how climate change can impact infectious diseases such as TB. The authors may want to expand upon this section.The connection between the 17 SDGs in Box 2, and the 11 SDGs likely to have a significant impact on the burden of TB in Section 5.1 could be more clear. Consider identifying in Box 2 (with asterisks, bold font, etc.) the subset of the 17 SDGs that will be described further in Section 5.1.Figure 1 is difficult to follow. It is unclear what the shapes, sizes and connections between the boxes indicate. All of the SDGs should be one size and shape. The other factors should be a different size or shape. It is unclear what the boxes grouped to the left of the diagram indicate. Also, several SDGs are missing numbers. There is no acknowledgment of strong NTP programs in this diagram. The diagram needs additional work.In section 6A of the paper, why are the SDGs (and their indicators) identified by WHO as relevant to TB different than those (both in number and type) than those listed in Section 5.1? For the 7 indicators listed in section 6A, and 7 indicators listed in section 6B, it would be helpful to include the SDG name for each rather than just the numerical reference (ex: No Poverty, Indicator 1).
  •  
  •  
  •  

There should be some definition of the Gini index for readers.

Given the multitude of possible interventions and the broad nature of the indicators it would seem that with the limited resources of many governments and NTPs, the authors would advocate for a prioritization of the SDGs that should be the focus for any individual country. One of the greatest difficulties with this manuscript is the impression that all SDGs should be addressed simultaneously for all countries. This seems unrealistic. Although it would be ideal to address all SDGs with the same vigor simultaneously, there needs to be some analysis or direction as to how countries (or NTPs) can prioritize some of these efforts. They are so broad and all encompassing, that the entire effort may potentially be perceived as overwhelming and perhaps paralyzing to a program for action. The authors should emphasize that countries may want to prioritize based on a criterion of the attributable SDG to the TB burden in a particular country. Providing and example or two of a possible approach in prioritization may be helpful.

Table 6 of possible interventions should be labeled as “Sample” interventions. Also, the table and interventions are listed as possible “community”-level interventions when many of the SDGs can (and should be) influenced by policy level interventions as well.  As such, the title should be amended to include the policy level changes as well. With the inclusion of policy there should be some mention of interventions for Climate Action-SDG-13. Also, in the table, (SDG-7) is referred to as Reduction in in-door air pollution—earlier in the manuscript this goal is referenced as SDG Goal 7: Sustainable and clean energy, which is broader than the in-door air pollution only. In fact, one of the possible interventions references larger environmental air pollution.

The manuscript would benefit from some discussion of other models or examples from other regions that may have used this SDG framework to address a health issue or topic. Is there any other literature on this broad approach that could be incorporated into the discussion and conclusion of this manuscript? The authors call on the need for research to parse out optimal mechanisms for collaboration and to measure the impact of such collaborations - so that the lessons learnt are disseminated widely. However, they fail to present any current literature or approaches from other disease areas or other regions that have had some success in this approach. It would be important to also acknowledge the importance of trying to ensure that any results of these interventions are not confounded by larger efforts to address TB by strengthening the core functions of NTPs.

The authors should also use the conclusion the reiterate that a strong and functioning NTP is an essential foundation needed to address TB and that the addition of addressing SDGs should not divert resources or energy away from the core public health components of (adequate diagnosis, treating, and prevention) of a successful public health program.

Reviewer 2 Report

1) It is a well written manuscript highlighting the need to look at social determinants of health for TB rather. However, these must be taken up in conjunction with a sound SEARCH, TREAT and PREVENT strategy.  I would suggest to authors to please add this in. 

2) The essential factor for TB infection and TB disease  Close contact with a person with bacteriologically positive pulmonary TB disease 

Recent unpublished data has shown that people with TB are identified in household of a bacteriological negative patients as well including those with EPTB. I would suggest to broaden this statement to all TB patients rather than only pulmonary patients. 

Author Response

Responses to comments from Reviewer 2

  • It is a well-written manuscript highlighting the need to look at social determinants of health for TB rather. However, these must be taken up in conjunction with a sound SEARCH, TREAT and PREVENT strategy.  I would suggest to authors to please add this in. 

Author’s Response: Many thanks for your encouraging words about our manuscript. We have highlighted that the efforts to address the socio-economic determinants should be undertaken along with the SEARCH, TREAT and PREVENT strategy. (manuscript line numbers 139-145)

  • The essential factor for TB infection and TB disease Close contact with a person with bacteriologically positive pulmonary TB disease. Recent unpublished data has shown that people with TB are identified in the household of a bacteriological negative patients as well including those with EPTB. I would suggest to broaden this statement to all TB patients rather than only pulmonary patients. 

Author’s response: We agree that there are instances of persons with nasopharyngeal and laryngeal TB (EPTB) spreading the disease.  Therefore we have modified the text to indicate that close contacts with persons with infectious TB is essential for the TB disease to occur. However, we feel that not all forms of EPTB are infectious (manuscript Revised Box 1 line number 110-111).

Round 2

Reviewer 1 Report

Thank you for addressing the items outlined in the original round of feedback. No additional edits requested.

Author Response

Reviewer’s comment: Thank you for addressing the items outlined in the original round of feedback. No additional edits requested.

Authors’ response: Thank you for indicating to us that our revisions are satisfactory. We wholeheartedly thank you for your constructive feedback on our manuscript. We have also done a spell check as indicated in your review report form.